# Diagnostic performance of chest computed tomography during the epidemic wave of COVID-19 varied as a function of time since the beginning of the confinement in France

Samia Boussouar[1], Mathilde Wagner[2,3], Victoria Donciu[2], Nicoletta Pasi[1], Joe Elie Salem[4], Raphaele Renard-Penna[2], Stéphane Marot[5], Yonathan Freund[6], Alban Redheuil[1], Olivier Lucidarme[2,3]*

1 LIB Biomedical Imaging Laboratory INSERM, CNRS, ICT Cardiothoracic Imaging Unit & Radiology Department, ICAN Institute of Cardiometabolism and Nutrition, Pitié-Salpêtrière Hospital (AP-HP), Sorbonne Université, Paris, France, 2 Department of Radiology, APHP, Sorbonne Universités, Paris, France, 3 Laboratoire d'Imagerie Biomédicale, CNRS, INSERM, APHP, Pitié-Salpêtrière Hospital, Sorbonne Universités, Paris, France, 4 Department of Pharmacology, CIC-1901, APHP, Sorbonne Universités, Paris, France, 5 Department of Virology, AP-HP, Pitié-Salpêtrière Hospital, INSERM 1136, Institut Pierre Louis d'Epidémiologie et de Santé Publique (iPLESP), Sorbonne Université, Paris, France, 6 Emergency Department, APHP, Pitié-Salpêtrière Hospital, Sorbonne Universités, Paris, France

* Olivier.lucidarme@aphp.fr

**Data Availability Statement:** Repository information for our data is: Lucidarme, Olivier

## Abstract

### Objective

To evaluate the diagnostic performance of the initial chest CT to diagnose COVID-19 related pneumonia in a French population of patients with respiratory symptoms according to the time from the onset of country-wide confinement to better understand what could be the role of the chest CT in the different phases of the epidemic.

### Material and method

Initial chest CT of 1064 patients with respiratory symptoms suspect of COVID-19 referred between March 18th, and May 12th 2020, were read according to a standardized procedure. The results of chest CTs were compared to the results of the RT-PCR.

### Results

546 (51%) patients were found to be positive for SARS-CoV2 at RT-PCR. The highest rate of positive RT-PCR was during the second week of confinement reaching 71.9%. After six weeks of confinement, the positive RT-PCR rate dropped significantly to 10.5% (p<0.001) and even 2.2% during the two last weeks. Overall, CT revealed patterns suggestive of COVID-19 in 603 patients (57%), whereas an alternative diagnosis was found in 246 patients (23%). CT was considered normal in 215 patients (20%) and inconclusive in 1 patient. The overall sensitivity of CT was 88%, specificity 76%, PPV 79%, and NPV 85%. At week-2, the same figures were 89%, 69%, 88% and 71% respectively and 60%, 84%, 30% and 95% respectively at week-6. At the end of confinement when the rate of positive PCR

(2020), "COVIDCT-PSL2", Mendeley Data, V1, doi: 10.17632/t2s38pzn66.1.

**Funding:** The authors received no specific funding for this work.

became extremely low the sensitivity, specificity, PPV and NPV of CT were 50%, 82%, 6% and 99% respectively.

## Conclusion

At the peak of the epidemic, chest CT had sufficiently high sensitivity and PPV to serve as a first-line positive diagnostic tool but at the end of the epidemic wave CT is more useful to exclude COVID-19 pneumonia.

## Introduction

On March 12th, 2020, the World Health Organization (WHO) declared COVID-19 to be a pandemic. After a rapid spread in Asian countries starting in December 2019, Europe was affected from the end of February 2020. The first French patient was officially diagnosed on February 19th, 2020 and very quickly the epidemic wave hit France with a peak during the months of March and April 2020. Strict nationwide confinement was ordered on March 17th, at this time the basic reproduction rate of the virus (R0) was 2.8 Simultaneously, dedicated COVID-19 patient pathways were organized throughout the healthcare system as well as specific imaging procedures and strategies in our institution, a prime referral center for COVID-19 patients in Paris, using chest computed tomography (CT) as a first line imaging modality for the management of symptomatic patients suspect of COVID-19. Ai et al. [1] reported early in February in a series of 1014 Chinese patients that chest CT had a 97% (CI$_{95\%}$: 95–98%) sensitivity and a 25% (CI$_{95\%}$: 22–30%) specificity for detecting COVID-19, whereas at that time real-time reverse transcriptase-polymerase chain reaction (RT-PCR), considered as the reference standard to diagnose SARS-CoV-2 infection had an imperfect sensitivity around 60–70% reported to be variable according to sample site [2]. Moreover, initial RT-PCR required long processing times and needed to be repeated over time. These figures were mostly confirmed in numerous other small Asian studies with sensitivities for CT ranging from 86 to 99% [3, 4] and specificities from 25 to 53% [5]. Chest CT was therefore recommended during the early phase of the epidemic to perform rapid triage for referral of patients to appropriate COVID-19 sectors to avoid emergency department overcrowding and initiate optimal therapy. On May 11th, the French government decided to begin a deconfinement process because of a dramatic decrease in viral spread with R0 estimation around 0.8. The role of chest CT in the deconfinement strategy remained in question. Indeed, a diagnostic test sensitivity and specificity are commonly believed not to vary with disease prevalence [6, 7] whereas positive predictive values are directly dependent on the prevalence of the disease and therefore cannot be directly transposed across different phases of the epidemic wave or across a country. Hence, the primary objective of our study was to evaluate the diagnostic performance of the initial chest CT to diagnose COVID-19 related pneumonia in a French population of consecutive patients with respiratory symptoms according to the time from the onset of confinement to take into account the variability of COVID-19 prevalence as a function of time and to better understand what could be the role of CT scan in different phases of the epidemic and potential future disease outbursts.

## Material and method

This retrospective study was approved by our institutional review board, the "Comité d'éthique de la Recherche-Sorbonne Université" and recorded as NCT04320017. According to the

French law concerning retrospective studies of medical records, the patient were informed by post-mail that anonymized data from their medical records would be reviewed within the framework of this retrospective study and that, in case of non-opposition their data would be included in the study. Fourteen patients expressed their opposition and were consequently not included, whereas the 1064 patients who did not express opposition were included. Data from their medical records was anonymized and then studied in this research.

## Patients

The study population consisted of consecutive adult patients referred to a large tertiary hospital between March 18[th] and May 12[th], 2020 for initial chest CT and nasopharyngeal RT-PCR test. Indications were: suspicion of COVID-19 with one or more of the following respiratory signs: dyspnea, polypnea, oxygen dependence. Patients suspected of having COVID-19 without respiratory signs were not admitted to the ER and did not have chest CT.

## CT exams

Detailed acquisition parameters are available as S1 File. In summary, all scans were performed on a dedicated scanner (Siemens Somaton Edge) either with or without iodine contrast injection according to the pretest risk of pulmonary embolism.

## Image analysis

A total of ten senior radiologists, aware of COVID-19 suspicion but unaware of RT-PCR results, read the CT exams in real time (prospectively). They used a standardized report which included typical, atypical and very atypical findings [8–10] listed in Table 1. Imaging findings used were very similar to the one described by Prokop et al. [11] as described in Covid-Rads which was published after the completion of this study. Pulmonary embolism was considered separately. Pleural effusion was considered neither in favor nor against the COVID-19 diagnosis. The conclusion of the report was therefore one of the following: 1) imaging patterns suggesting the presence of COVID-19; 2) imaging patterns suggesting an alternative diagnosis; 3) imaging patterns suggesting a combination of COVID-19 with underlying lung disease; 4) CT considered normal.

## RT-PCR

Nasopharyngeal swab samples were also performed to allow detection of SARS-CoV-2 RNA (S2 File). In the case of a first negative RT-PCR test, repeat tests were performed. For patients

**Table 1. Imaging findings used in the standardized reports.**

| Typical findings of Covid 19 | Atypical findings | In favor of another diagnosis |
|---|---|---|
| Multifocal ground glass opacities (GGO) | Lymphadenopathies | Tree in bud micronodules, bronchiolitis |
| Peripheral and basal distribution of GGO | systematized condensation with aerial bronchogram | Micronodules with lymphatic or random distribution |
| Unsharp demarcation | | Mass and nodules |
| Crazy paving | | Mosaic perfusion |
| Association of ground glass and consolidations | | Cavitations |
| Consolidations, rather linear and rather located at the periphery of the lung | | Central and peribronchovascular distribution of GGO |
| | | Calcifications |

with multiple RT-PCR tests, the diagnosis of COVID-19 was confirmed when at least one test was positive within 15 days of CT.

## Epidemiological data

The exact prevalence of the disease in the Paris area was not known at the time of admission, thus we collected, on a weekly basis, the absolute number of COVID-19 patients admitted into hospitals in the Paris region (Ile de France) and in the city of Paris obtained from government statistics (https://www.gouvernement.fr/info-coronavirus/carte-et-donnees) between March 18th and May 12th. We also collected the estimation of R0 values every weeks for the same period of time (from https://www.qualitiso.com/coronavirus-analyse-des-risques/).

## Statistical analysis

For more detail see S3 File. Using RT-PCR results as reference, the sensitivity, specificity, positive predictive value (PPV), negative predictive value (NPV) and overall accuracy of chest CT imaging were calculated. The performance measures of chest CT for identifying COVID-19 as a function of time elapsed from the onset of confinement was also calculated. The observation period of this study lasted 8 weeks and was split into 8 weekly periods from the first day to the last day of confinement. Percentage of positive PCR over the 8 weeks, as well as the accuracy of chest CT over the 8 weeks were compared using the Chi-square test.

## Results

Between March 18, and May 12, 2020, 1787 CT scans were performed. As summarized in Fig 1, 1064 patients, 583 males (55%) and 481 females (45%), with a mean age of 64 years (range

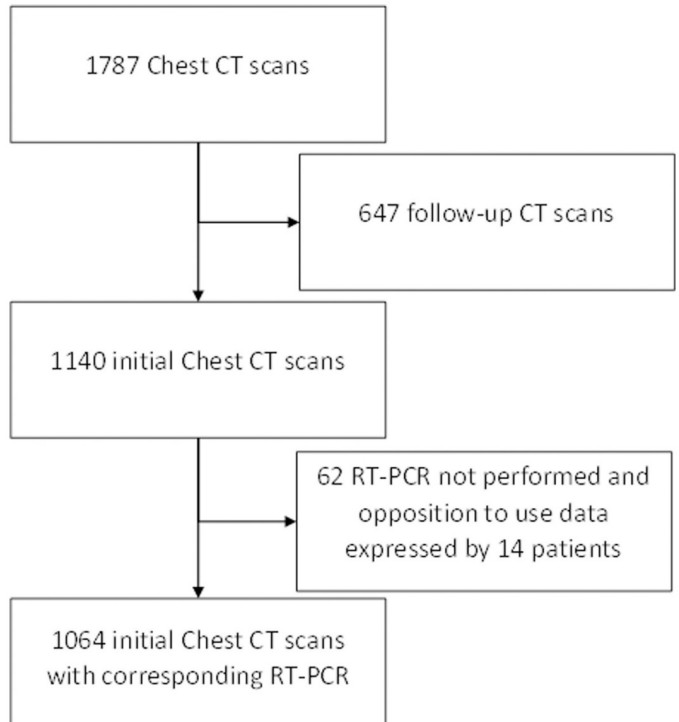

**Fig 1. Study flow chart.**

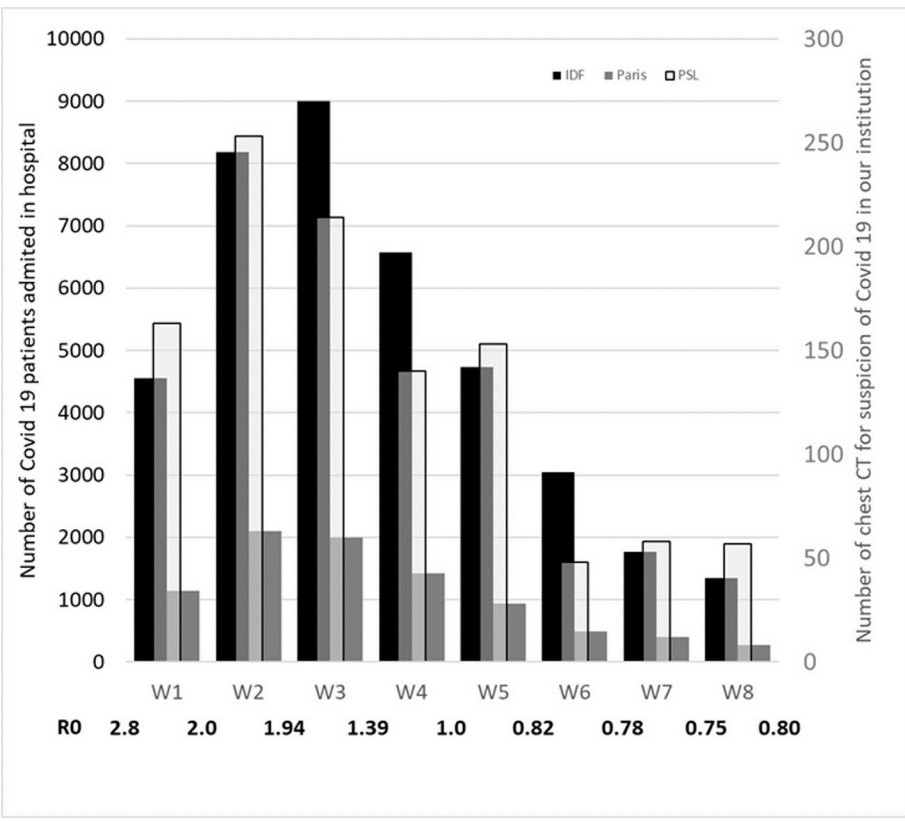

**Fig 2. R0, number of COVID 19 patients admitted in the hospital and number of positive RT-PCR in our institution.** Number of COVID-19 patients admitted in the hospital in the whole area around Paris (Ile de France: IDF) and in the city of Paris (Paris) (left axis) and number of positive RT-PCR in our institution (PSL) (right axis) and value of R0 as a function of weeks elapsed since the beginning of the confinement W1 = 18–24 of March, W2 = 25–31 of March, W3 = 01–07 of April, W4 = 08–14 of April, W5 = 15–21 of April, W6 = 22–28 of April, W7 = 29 of April-5 of May, W8 = 06–12 of May.

18–100) and a mean weight (when known, n = 465) of 75kg (range 32–183) met the inclusion criteria (Fig 1). The time elapsed since the onset of respiratory symptoms averaged 48 hours (range 2–168 h).

The first RT-PCR was positive in 516 (48%) of cases. Considering subsequent RT-PCRs, 546 (51%) patients were ultimately found to be positive for SARS-CoV2. Fig 2 shows the absolute number of COVID-19 patients admitted in the Paris Metropolitan Area and Region hospitals and the absolute number of positive RT-PCR in our institution as a function of weeks elapsed since the beginning of the confinement Values of R0 estimated at the beginning of each week and at the end of W8 are also reported on Fig 2.

Table 2 shows the concurrent number and percentage of positive CT and RT-PCR for COVID-19 as a function of weeks. The highest positive RT-PCR rate was recorded in our institution during the second week of confinement (25–31 of March) with 71.9%. After six weeks of confinement, the positive RT-PCR rate dropped significantly to 10.5% (p<0.001) and then 4.7% at week 7 and 0% at week 8. CT revealed imaging patterns suggesting the diagnosis of COVID-19, in 603 patients (57%), either alone (520/603, 86%) or in combination with another lung disease (83/603, 14%) (Fig 3). The most common CT patterns for COVID-19 pneumonia seen in this study were ground glass opacities, crazy paving and consolidation mostly distributed in subpleural regions and involving all lobes (Fig 3). Imaging patterns in favor of another

**Table 2. Contingency table of the results of CT and RT-PCR for COVID-19 as a function of weeks from country-wide confinement.**

| | | | RT-PCR | | |
| --- | --- | --- | --- | --- | --- |
| | | | **Negative** | **Positive** | **Total** |
| Chest CT | W1 | Negative | 35 (21) | 10 (6) | *45 (28)* |
| | | Positive | 17 (10) | 101 (62) | *118 (72)* |
| | | *Total* | *52 (32)* | *111 (68)* | *163 (100)* |
| | W2 | negative | 49 (19) | 20 (8) | *69 (27)* |
| | | positive | 22 (9) | 162 (64) | *184 (73)* |
| | | *total* | *71 (28)* | *182 (72)* | *253 (100)* |
| | W3 | negative | 60 (28) | 17 (8) | *77 (36)* |
| | | positive | 28 (13) | 109 (51) | *137 (64)* |
| | | *total* | *88 (41)* | *126 (59)* | *214 (100)* |
| | W4 | negative | 55 (39) | 8 (6) | *63 (45)* |
| | | positive | 22 (16) | 55 (39) | *77 (55)* |
| | | *total* | *77 (55)* | *63 (45)* | *140 (100)* |
| | W5 | negative | 84 (55) | 9 (6) | *93 (61)* |
| | | positive | 12 (8) | 48 (31) | *60 (39)* |
| | | *total* | *96 (63)* | *57 (37)* | *153 (100)* |
| | W6 | negative | 36 (75) | 2 (4) | *38 (79)* |
| | | positive | 7 (15) | 3 (6) | *10 (21)* |
| | | *total* | *43 (90)* | *5 (10)* | *48 (100)* |
| | W7 | negative | 34 (77) | 1 (2) | *35 (80)* |
| | | positive | 8 (18) | 1 (2) | *9 (20)* |
| | | *total* | *42 (95)* | *2 (5)* | *44 (100)* |
| | W8 | negative | 40 (83) | 0 (0) | *40 (83)* |
| | | positive | 8 (17) | 0 (0) | *8 (17)* |
| | | *total* | *48 (100)* | *0 (0)* | *48 (100)* |
| | **total 8 weeks** | **negative** | **393 (37)** | **67 (6)** | ***460 (43)*** |
| | | **positive** | **124 (12)** | **479 (45)** | ***603 (57)*** |
| | | ***total*** | ***517(49)*** | ***546 (51)*** | ***1063 (100)*** |

Note: The table reports the number of patients n and percentage (%). W1 = 18–24 of March, W2 = 25–31 of March, W3 = 01–07 of April, W4 = 08–14 of April, W5 = 15–21 of April, W6 = 22–28 of April, W7 = 29 of April-5 of May, W8 = 06–12 of May

diagnosis was found in 246 patients (23%) (Fig 3) while CT was considered as normal in 215 patients (20%). One CT was unconclusive.

Concerning the 30 cases with initially negative RT-PCR subsequently positive, 18 initial CT scans were positive for COVID-19 (60%). Using at least one positive RT-PCR as a reference standard, the overall sensitivity of CT was 88%, overall specificity 76%, PPV 79%, and NPV 85%. Table 3 shows the accuracies of CTs obtained during the 8 weeks of the study. During the second week of confinement, the sensitivity, specificity, PPV and NPV of CT were 89%, 69%, 88% and 71% respectively. After six weeks of confinement, the same figures were 60%, 84%, 30% and 95% respectively. At the end of confinement, when the rate of positive PCR became extremely low, the sensitivity, specificity, PPV and NPV of CT were 50%, 82%, 6% and 99% respectively. Overall, accuracy remained stable over the 8 weeks (p = 0.42).

## Discussion

Our study found that, during the national spring confinement, the PPV of CT to diagnose COVID-19 dramatically dropped simultaneously with the decrease of R0 and of the number of

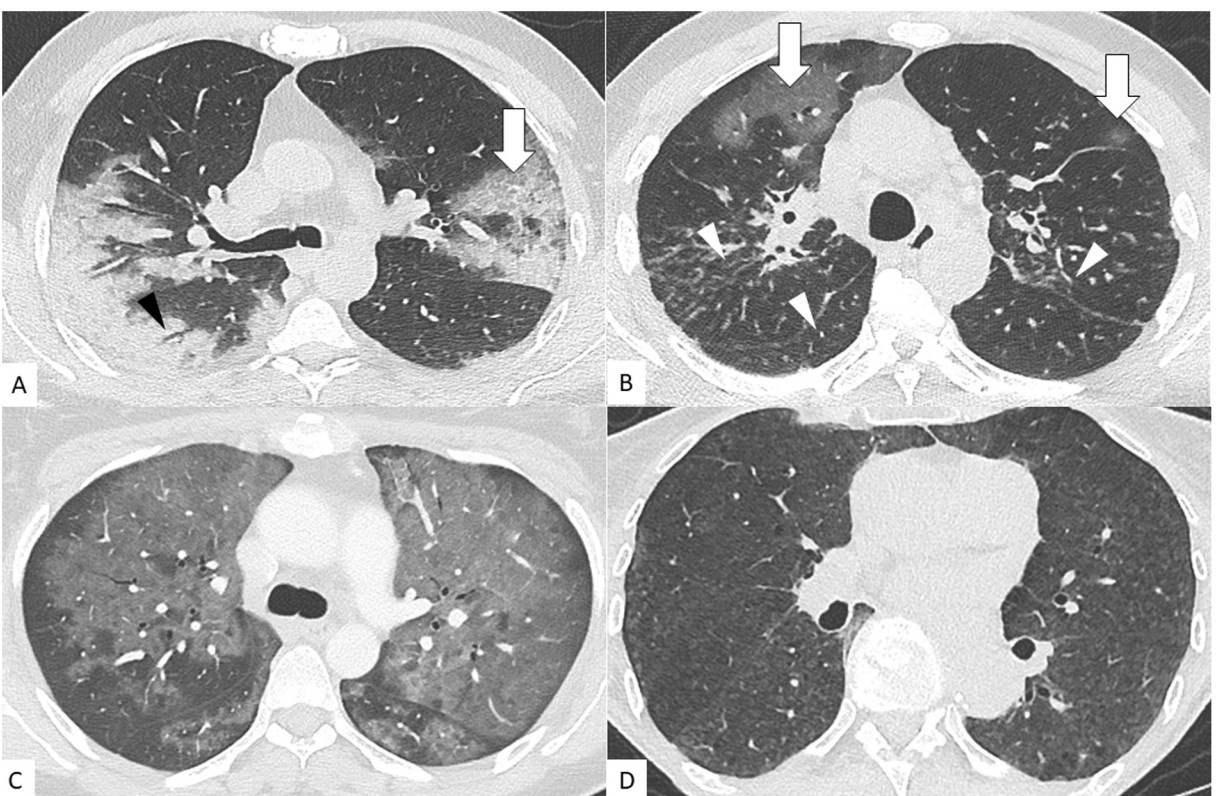

**Fig 3. Lung CT patterns found in patients with and without COVID-19 pneumonia.** A) Unenhanced chest CT images of a 62-year-old man with fever and dyspnea revealing one of the typical CT patterns for COVID-19 pneumonia. Axial images show ground glass opacities, crazy paving pattern (arrows) and consolidation with air bronchogramm (arrowhead) mostly distributed in subpleural regions and involving all lobes. RT-PCR here was positive (suggesting true positive diagnosis of CT). B) Unenhanced chest CT images of a 56-year-old man with COVID-19 pneumonia and sarcoidosis. Axial CT images revealed bilateral ground glass opacities (arrows) and perilymphatic irregular nodular thickening in an upper/mid lung distribution (arrowheads). RT PCR was positive (suggesting true positive diagnosis of CT). C) Enhanced chest CT images of a 37-year-old woman with dyspnea and fever revealing *Pneumocystis Jiroveci* infection. Axial CT images shows central diffuse GGO bilateral falsely considered suggesting of COVID-19. RT-PCR was here negative (suggesting false positive diagnosis of CT). D) Unenhanced chest CT images of a 67-year-old woman with dyspnea for few days revealing a hypersensitivity pneumonitis. Axial CT images shows homogeneous GGO bilateral and symmetric with a bronchovascular distribution. CT was considered suggesting another diagnosis than COVID-19. RT-PCR was here negative (suggesting true negative diagnosis of CT).

**Table 3. Diagnostic performance of chest CT as a function of weeks elapsed since the beginning of the country-wide confinement.**

|           | Sensitivity | Specificity | PPV | NPV |
|-----------|-------------|-------------|-----|-----|
| **W1**    | 91          | 67          | 86  | 78  |
| **W2**    | 89          | 69          | 88  | 71  |
| **W3**    | 87          | 68          | 80  | 78  |
| **W4**    | 87          | 71          | 71  | 87  |
| **W5**    | 84          | 88          | 80  | 90  |
| **W6**    | 60          | 84          | 30  | 95  |
| **W7 + W8** | 50        | 82          | 6   | 99  |
| **Overall** | **88**    | **76**      | **79** | **85** |

Note: PPV, NPV: positive and negative predictive values. Reported numbers are percentages, %. W1 = 18–24 of March, W2 = 25–31 of March, W3 = 01–07 of April, W4 = 08–14 of April, W5 = 15–21 of April, W6 = 22–28 of April, W7 = 29 of April-5 of May, W8 = 06–12 of May. W7 and W8 were merged because of the very low number of positive RT-PCR during these two weeks.

hospitalized patients in the area which are both indirect markers of the local disease prevalence during confinement."Six weeks after the onset of confinement PPV was only 30%. This value dropped even lower at 6% during the two last weeks of confinement. This low value is related to the rise of false positive results from 9% during the epidemic peak (week 2), to 15% in week 6 and 17% in week 8 of confinement. The CT reader's expectation, which is influenced by the supposed prevalence of the disease, plays a major role [12, 13]. During the epidemic, radiologists tend to conclude that any ground glass opacity or crazy paving or peripheral linear zone of condensation suggests the presence of COVID-19. This tendency persists even during the declining phase of the epidemic but these imaging features are not specific of COVID-19 pneumonia but shared with other viral etiologies as well as with hypersensitivity pneumonia [14]. This increases the false positive rate, especially as the prevalence decreases rapidly. Consequently, although the initial CT is a valuable tool for managing patients at the peak of the epidemic (in our study the two first weeks of confinement when R0 was greater than 2.0) without waiting for RT-PCR results, this is no longer the case once the epidemic wave has been contained by confinement. Extensive use of CT as an initial tool to manage new patients suspected of COVID-19 at the beginning of the deconfinement period could result in a significant number of false positive results leading to poor patient management including mistakenly placing the patient in a COVID-19 pathway.

The rise in false positive results should also induce a decrease of the specificity of CT and should have no consequence on sensitivity and negative predictive values. However, in this study we found the opposite, sensitivity decreased while specificity increased as COVID-19 prevalence decreased over time. Leeflang *et al.* [13] reported that sensitivity and specificity can be either higher or lower with lower prevalence. In our study, the increase in specificity with time elapsed since the beginning of confinement, despite a higher number of false positive results, is necessarily explained by an even greater increase in true negatives (19% at week 2, 75% at week 6 and 83% at week 8 of confinement), which can be mechanically explained by the decrease in RT-PCR positive cases. The decrease of positive cases was noteworthy particularly after W5 and was a result of the combination of two phenomena: 1) Although Fig 2 shows a similar evolution profile between the number of hospitalizations in the region around Paris and in the city of Paris and the number of patients referred to our CT for Covid19 suspicion and respiratory symptoms, it appeared that this number fell more rapidly between W5 and W6 (-67%) than the number of new patients hospitalized, which only decreased by 47% in Paris and 36% in the region over the same period of time. This rapid decrease had no explanation but does not seem to correspond to a statistical variation because at W7 and W8 the number of patients referred to our CT for respiratory symptoms remained similar between 44% and 48% regardless of the trend towards a pursuit of the decrease of R0 and in the number of patients hospitalized for Covid19 in the region and the city of Paris. 2) at the same time, the proportion of positive PCR among patients referred to the scanner with respiratory symptoms went from 37% at W5 to 10% at W6 (-73%) and even 4.7% at W7 (-90%) and 0% at W8. This can be explained by a strong efficacy of the nationwide confinement combined with a return to the emergency department of patients with other respiratory pathologies (COPD, asthma, bacterial pneumonia,. . .), a return that became more pronounced as the epidemic was losing ground. These patients with other respiratory diseases were considered to be clinically suspect for COVID19 but confirmed to be true negatives of CT and likely caused a significant drop in the percentage of positive PCRs.

This also explains the increase of the NPV as a function of time during the confinement. Finally, the decrease of sensitivity observed when prevalence decreased is explained by both drop down of FN and TP results. The TP rate was 64% at week 2,6% at week 6 and around 1% the two last weeks of confinement. These results show that in deconfinement period CT scan is

more useful to exclude COVID-19 than to diagnose it in patients with respiratory symptoms. Our study has certain limitations: RT-PCR on nasophynrageal swab samples, considered as the reference, had a significant number of false negative results which could be explained by several factors such as an inappropriate sampling technique, an inadequate transportation procedure or by insufficient viral load according to the natural history of the COVID-19 with a shift of the preferential viral replication site from the upper respiratory tract to the lower respiratory tract during the course of the infection [15]. Repeated testing could be necessary to avoid misdiagnosis and improve RT-PCR sensitivity; therefore, false positive results of CT were overestimated, leading to an underestimation of specificity; however, since we took into account at least one repeat RT-PCR positive result as the final confirmatory diagnosis, we probably minimized this bias. We did not re-analyze the chest CT scans and only considered the result of the report done at the time of the CT. We therefore merged the individual performances of ten different senior radiologists with unequal experience of lung disease. However, the resulting overall accuracy of the chest CT scan was more reflective of the reality of a night shift and, in addition, two chest radiologists (SB and VD) were always available to analyze difficult imaging results before final report validation. In addition, the duty distribution between each radiologist was even, in other words there was no reading bias by radiologists with higher sensitivity at the beginning of the wave and by less sensitive radiologists at the end of the wave. We did not consider in the conclusion of the chest CT report an "indeterminate appearance" as recommended by the RSNA expert consensus statement [10]. Therefore we may have misclassified some "inderterminate" exams probably leading to an excess of false negatives at the peak of epidemic and to an excess of false positives at the end of the study. However, we had made the choice to conclude in a binary manner in favor or not in favor of COVID-19 in order to help clinicians for early triage avoiding the "third" choice "indeterminate" that is usually often use when made available. Finally, we did not use CT as a screening tool for asymptomatic patients; our results apply consequently only to the population of patients with respiratory symptoms.

## Conclusion

In patients with respiratory symptoms, diagnostic performance by CT for COVID-19 varied during the epidemic wave. At the peak of the epidemic, Chest CT had sufficiently high sensitivity and PPV to serve as a first-line diagnostic and triage tool without waiting for RT-PCR results. Conversely, at the end of the epidemic wave and beginning of deconfinement period, when the prevalence of COVID-19 becomes low, CT is more useful for excluding COVID-19 and making alternate diagnoses.

## Supporting information

**S1 File. CT scanning.**
(DOCX)

**S2 File. RT PCR.**
(DOCX)

**S3 File. Statistical analysis.**
(DOCX)

## Acknowledgments

Emina Arsovic RES, Laure Gracia RES, Domitille Droz-Bartholet RES, Benjamin Dray RES, Antoine Pelcat RES, Laurene Aupin RES, Camille Thoumin RES, Amandine Chabernaud

Negrier RES, Alexandre Héraud RES, Josua Vegas RES, Laura Medioni RES, Dan Toledano MD

## Author Contributions

**Conceptualization:** Alban Redheuil, Olivier Lucidarme.

**Data curation:** Samia Boussouar, Mathilde Wagner, Joe Elie Salem, Raphaele Renard-Penna, Stéphane Marot, Alban Redheuil, Olivier Lucidarme.

**Formal analysis:** Samia Boussouar, Mathilde Wagner, Victoria Donciu, Raphaele Renard-Penna, Alban Redheuil, Olivier Lucidarme.

**Investigation:** Samia Boussouar, Victoria Donciu, Nicoletta Pasi, Stéphane Marot, Yonathan Freund, Alban Redheuil.

**Methodology:** Samia Boussouar, Mathilde Wagner, Joe Elie Salem, Yonathan Freund.

**Project administration:** Joe Elie Salem, Alban Redheuil.

**Resources:** Stéphane Marot.

**Software:** Mathilde Wagner.

**Supervision:** Olivier Lucidarme.

**Validation:** Alban Redheuil, Olivier Lucidarme.

**Writing – original draft:** Samia Boussouar, Olivier Lucidarme.

**Writing – review & editing:** Mathilde Wagner, Victoria Donciu, Alban Redheuil, Olivier Lucidarme.

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
