## [Decision Letter · Decision Letter 0]

17 Sep 2020

PONE-D-20-23438

Diagnostic Performance of Chest Computed Tomography during the Epidemic Wave of COVID-19 Varies as a Function of Time since the Beginning of the Confinement in France

PLOS ONE

Dear Dr. Lucidarme,

Thank you for submitting your manuscript to PLOS ONE. After careful consideration, we feel that it has merit but does not fully meet PLOS ONE’s publication criteria as it currently stands. Therefore, we invite you to submit a revised version of the manuscript that addresses the points raised during the review process.

We look forward to receiving your revised manuscript.

Kind regards,

Ming-Ching Lee

Academic Editor

PLOS ONE

Journal Requirements:

2. In ethics statement in the manuscript and in the online submission form, please provide additional information about the patient records used in your retrospective study. Specifically, please ensure that you have discussed whether all data were fully anonymized before you accessed them and/or whether the IRB or ethics committee waived the requirement for informed consent. If patients provided informed written consent to have data from their medical records used in research, please include this information.

5. We note you have included a table to which you do not refer in the text of your manuscript. Please ensure that you refer to Table 3 in your text; if accepted, production will need this reference to link the reader to the Table.

Reviewers' comments:

Reviewer's Responses to Questions

**Comments to the Author**

1. Is the manuscript technically sound, and do the data support the conclusions?

Reviewer #1: Partly

Reviewer #2: Yes

2. Has the statistical analysis been performed appropriately and rigorously? 

Reviewer #1: Yes

Reviewer #2: Yes

3. Have the authors made all data underlying the findings in their manuscript fully available?

Reviewer #1: Yes

Reviewer #2: Yes

4. Is the manuscript presented in an intelligible fashion and written in standard English?

Reviewer #1: Yes

Reviewer #2: Yes

5. Review Comments to the Author

Reviewer #1: In the study the authors described the impact of disease prevalence on the diagnostic performance of chest CT during the pandemic COVID-19 period. The data was analyzed reasonably. I have one major question and two minors for the authors.

1.The case numbers decreased substantially from week 1 to week 6. However, the confirmed COVID-19 cases decreased more than 90% within one week (from 57 cases in week 5 to 5 cases in week 6), which was relatively a sudden drop as compared with data in the previous weeks. This may raise the concern of sampling bias in week 6 and affect the final results of your study. Would you compare your data to the decline rate of prevalence in your region during the same period?

2.In the context of Results, the sensitivity, specificity, PPV and NPV in week 2 and week 6 were not compatible with those in table 3. Please clarify which is correct.

3.Please review your reference and make corrections as errors were noted (at least ref. 5: wrong authors and wrong DOI, and ref. 10: wrong authors and no DOI).

Reviewer #2: This manuscript concerns an important topic to validate the role of chest CT scan during the COVID-19 endemic. The study design is reasonable to practice and the research outcomes are clear. I think it can provide useful information to clinicians.

Some questions as following

Question 1: What is the most common presentation of CT findings regarding the COVID-19 pneumonia in the current study?

Question 2: As your findings in the current study, the interpretation of CT will be influenced by the supposed prevalence of the disease. What time is the optimal time to use CT scan as the initial screening tool for symptomatic patients who are suspected to have COVID-19 infection? For example, R0 value? in the period of local transmission or community spread?

6. PLOS authors have the option to publish the peer review history of their article (what does this mean?). If published, this will include your full peer review and any attached files.

Reviewer #1: **Yes: **Shaw-Woei Leu

Reviewer #2: **Yes: **Dr. Pin-Kuei Fu. M.D., Ph.D

---

## [Author Response · Author response to Decision Letter 0]

20 Oct 2020

Journal Requirements:

E1. Please ensure that your manuscript meets PLOS ONE's style requirements, including those for file naming. The PLOS ONE style templates can be found at

https://clicktime.symantec.com/3YVDg2FUmJPdJafS3bt6WuN6H2?u=https%3A%2F%2Fjournals.plos.org%2Fplosone%2Fs%2Ffile%3Fid%3DwjVg%2FPLOSOne_formatting_sample_main_body.pdf and

https://clicktime.symantec.com/3V18fZ6tFtcSdAr3Y2XCj8A6H2?u=https%3A%2F%2Fjournals.plos.org%2Fplosone%2Fs%2Ffile%3Fid%3Dba62%2FPLOSOne_formatting_sample_title_authors_affiliations.pdf

Done. The reviewed manuscript fulfills PLOS ONE style templates.

E2. In ethics statement in the manuscript and in the online submission form, please provide additional information about the patient records used in your retrospective study. Specifically, please ensure that you have discussed whether all data were fully anonymized before you accessed them and/or whether the IRB or ethics committee waived the requirement for informed consent. If patients provided informed written consent to have data from their medical records used in research, please include this information.

According to the French law concerning retrospective studies of medical records (Jardé law), the patient must be informed by post-mail that anonymized data from their medical records will be reviewed within the framework of a given retrospective study. Patients are included only if they do not oppose the use of their anonymized medical records for research purposes. For this study, all patients (or their families) received a letter of information. 14 expressed their opposition and were therefore not included. 1064 patients who did not express opposition were thus included and anonymized prior to study in this research. 

In order to answer the first comment of R1 we analyzed also the medical records of patients referred to CT for suspicion of COVID19 with respiratory symptoms at week 7 and 8. Hence we included in this revised manuscript an additional set of 92 patients to whom we also sent a letter of information mid-September. None of them expressed an opposition (Cf. above).

The Ethics Committee approval was referenced CER-Sorbonne Université 2020-014. We know provide in the revised manuscript the NCT number: NCT04320017.

E3. Thank you for stating the following financial disclosure:

a. Please clarify the sources of funding (financial or material support) for your study. List the grants or organizations that supported your study, including funding received from your institution.

There was no specific source of funding for this study, the authors are all paid employees of Academic Institutions the AP-HP “Assistance-Publique-Hôpitaux de Paris” and/or “Sorbonne University” and the material support (computer, software, database) was also provided by these 2 institutions for the routine clinical practice and routine research work.

d. If you did not receive any funding for this study, please state: “The authors received no specific funding for this work.”

The authors received no specific funding for this work.

E4. We note that you have stated that you will provide repository information for your data at acceptance. Should your manuscript be accepted for publication, we will hold it until you provide the relevant accession numbers or DOIs necessary to access your data. If you wish to make changes to your Data Availability statement, please describe these changes in your cover letter and we will update your Data Availability statement to reflect the information you provide.

Repository information for our data is : Lucidarme, Olivier (2020), “COVIDCT-PSL2”, Mendeley Data, V1, doi: 10.17632/t2s38pzn66.1 

E5. We note you have included a table to which you do not refer in the text of your manuscript. Please ensure that you refer to Table 3 in your text; if accepted, production will need this reference to link the reader to the Table.

To answer R1.1 we changed tables and figures and we renumbered accordingly in the revised manuscript.

Responses to Reviewers

Reviewer #1: In the study the authors described the impact of disease prevalence on the diagnostic performance of chest CT during the pandemic COVID-19 period. The data was analyzed reasonably. I have one major question and two minors for the authors.

R1.1. The case numbers decreased substantially from week 1 to week 6. However, the confirmed COVID-19 cases decreased more than 90% within one week (from 57 cases in week 5 to 5 cases in week 6), which was relatively a sudden drop as compared with data in the previous weeks. This may raise the concern of sampling bias in week 6 and affect the final results of your study. Would you compare your data to the decline rate of prevalence in your region during the same period?

Thank you for this important question which allowed us to deepen our reflection by introducing a comparative analysis with the evolution of the epidemic in both the Paris area, the Ile de France (IDF) Region and the city of Paris. In the revised manuscript, we removed Figures 2 and 4, which described the number of positive PCR in our institution and the accuracy of CTs which were redundant with table 3. We added a new figure 2, with a double-y axis figure that simultaneously displays the number of COVID 19 patients admitted into hospitals in the Paris region (IDF) and the city of Paris obtained from government statistics (https://www.gouvernement.fr/info-coronavirus/carte-et-donnees) because the exact prevalence of the disease was not known. In addition we added data from two additional weeks (Week 7 & week 8) in order to show that results in W6 were not pitfalls but was part of a general trend and we have commented on these results in the new discussion section as follows.

. The decrease of positive cases was noteworthy particularly after W5 and was a result of the combination of two phenomena: 1) Although Figure 2 shows a similar evolution profile between the number of hospitalizations in the region around Paris and in the city of Paris and the number of patients referred to our CT for Covid19 suspicion and respiratory symptoms, it appeared that this number fell more rapidly between W5 and W6 (-67%) than the number of new patients hospitalized, which only decreased by 47% in Paris and 36% in the region over the same period of time. This rapid decrease had no explanation but does not seem to correspond to a statistical variation because at W7 and W8 the number of patients referred to our CT for respiratory symptoms remained similar between 44% and 48% regardless of the trend towards a pursuit of the decrease in the number of patients hospitalized for Covid19 in the region and the city of Paris. 

2) at the same time, the proportion of positive PCR among patients referred to the scanner with respiratory symptoms went from 37% at W5 to 10% at W6 (-73%) and even 5% at W7 (-90%) and 0% at W8. This can be explained by a strong efficacy of the nationwide confinement combined with a return to the emergency department of patients with other respiratory pathologies (COPD, asthma, bacterial pneumonia,...), a return that became more pronounced as the epidemic was losing ground. These patients with other respiratory diseases were considered to be clinically suspect for COVID-19 but were eventually confirmed to be true negatives of CT and likely caused a significant drop in the percentage of positive PCRs.

We modified accordingly the flow chart and the results displayed in tables 2 &3 to take into account results obtained at W7 and W8 and we added the following paragraph into the material and method section

“Epidemiological additional data 

The exact prevalence of the disease in the Paris area was not known at the time of admission, thus we collected, on a weekly basis, the absolute number of COVID-19 patients admitted into hospitals in the Paris region (Ile de France) and in the city of Paris obtained from government statistics (https://www.gouvernement.fr/info-coronavirus/carte-et-donnees) between March 18th and May 12th. We also collected the estimation of R0 values every weeks for the same period of time (from https://www.qualitiso.com/coronavirus-analyse-des-risques/). 

R1.2.In the context of Results, the sensitivity, specificity, PPV and NPV in week 2 and week 6 were not compatible with those in table 3. Please clarify which is correct.

Thanks for identifying this mistake. All calculations were redone and modifications made accordingly to the revised manuscript.

R1.3. Please review your reference and make corrections as errors were noted (at least ref. 5: wrong authors and wrong DOI, and ref. 10: wrong authors and no DOI).

Our apologies for these mistakes. We double checked the references.

Reviewer #2: This manuscript concerns an important topic to validate the role of chest CT scan during the COVID-19 endemic. The study design is reasonable to practice and the research outcomes are clear. I think it can provide useful information to clinicians.

Some questions as following

R2.1: What is the most common presentation of CT findings regarding the COVID-19 pneumonia in the current study?

To answer this question we added the following sentence in the result section : “The most common CT patterns for COVID-19 pneumonia seen in this study were ground glass opacities, crazy paving and consolidation mostly distributed in subpleural regions and involving all lobes (figure 3a).” 

These patterns are illustrated in Figure 3.

Question 2: As your findings in the current study, the interpretation of CT will be influenced by the supposed prevalence of the disease. What time is the optimal time to use CT scan as the initial screening tool for symptomatic patients who are suspected to have COVID-19 infection? For example, R0 value? in the period of local transmission or community spread?

According to Table 3, the optimal time to use CT scans as initial screening tools for symptomatic patient was at the beginning of the confinement period (W1 = 18-24 of March or W2 = 25-31 of March). This was also probably the time of the highest prevalence however its value was not known due to a lack of RT PCR tests in the population. The local prevalence is the key regardless of local transmission or a community spread once again unpredictable. To answer this comment, we now provide additional available information about the number of patients (precisely known) admitted in hospital in the Paris-region (see also R1.1) and in the city of Paris as well as the estimation of R0 as a function of time in the figure 2 and added the following sentences in the discussion section:

“Our study found that the PPV of CT to diagnose COVID-19 dramatically dropped simultaneously with decrease during confinement of R0 and of the number of hospitalized patients in the area which are both indirect markers of the local disease prevalence during confinement.”

“CT is a valuable tool for managing patients at the peak of the epidemic (in our study the two first weeks of confinement when R0 was greater than 2.0) without waiting for RT-PCR results, this is no longer the case once the epidemic wave has been contained by confinement.”

6. PLOS authors have the option to publish the peer review history of their article (what does this mean?). If published, this will include your full peer review and any attached files.

We do agree

---

## [Decision Letter · Decision Letter 1]

11 Nov 2020

Diagnostic Performance of Chest Computed Tomography during the Epidemic Wave of COVID-19 Varied as a Function of Time since the Beginning of the Confinement in France

PONE-D-20-23438R1

Dear Dr. Lucidarme,

We’re pleased to inform you that your manuscript has been judged scientifically suitable for publication and will be formally accepted for publication once it meets all outstanding technical requirements.

Kind regards,

Ming-Ching Lee

Academic Editor

PLOS ONE

Additional Editor Comments (optional):

Reviewers' comments:

Reviewer's Responses to Questions

**Comments to the Author**

1. If the authors have adequately addressed your comments raised in a previous round of review and you feel that this manuscript is now acceptable for publication, you may indicate that here to bypass the “Comments to the Author” section, enter your conflict of interest statement in the “Confidential to Editor” section, and submit your "Accept" recommendation.

Reviewer #1: All comments have been addressed

2. Is the manuscript technically sound, and do the data support the conclusions?

Reviewer #1: Yes

3. Has the statistical analysis been performed appropriately and rigorously? 

Reviewer #1: Yes

4. Have the authors made all data underlying the findings in their manuscript fully available?

Reviewer #1: Yes

5. Is the manuscript presented in an intelligible fashion and written in standard English?

Reviewer #1: Yes

6. Review Comments to the Author

Reviewer #1: (No Response)

7. PLOS authors have the option to publish the peer review history of their article (what does this mean?). If published, this will include your full peer review and any attached files.

Reviewer #1: **Yes: **Shaw-Woei Leu

---

## [Editor Report · Acceptance letter]

13 Nov 2020

PONE-D-20-23438R1 

Diagnostic performance of chest computed tomography during the epidemic wave of covid-19 varied as a function of time since the beginning of the confinement in france 

Dear Dr. Lucidarme:

I'm pleased to inform you that your manuscript has been deemed suitable for publication in PLOS ONE. Congratulations! Your manuscript is now with our production department. 

Kind regards, 

on behalf of

Dr. Ming-Ching Lee 

Academic Editor

PLOS ONE